# What Do We Know about Dog Owners? Exploring Associations between Pre-Purchase Behaviours, Knowledge and Understanding, Ownership Practices, and Dog Welfare

**DOI:** 10.3390/ani14030396

**Published:** 2024-01-25

**Authors:** Izzie Philpotts, Emily J. Blackwell, Justin Dillon, Emma Tipton, Nicola J. Rooney

**Affiliations:** 1Animal Behaviour and Welfare Group, Bristol Veterinary School, University of Bristol, Bristol BS40 5DU, UK; emily.blackwell@bristol.ac.uk (E.J.B.); nicola.rooney@bristol.ac.uk (N.J.R.); 2School of Sport, Rehabilitation and Exercise Sciences, University of Essex, Colchester CO4 3SQ, UK; 3IOE, UCL’s Faculty of Education and Society, University College London, London WC1H 0AL, UK; justin.dillon@ucl.ac.uk; 4PDSA, Whitechapel Way, Priorslee, Telford TF2 9PQ, UK; tipton.emma@pdsa.org.uk

**Keywords:** dogs, dog acquisition, pre-purchase, owner behaviours, owner knowledge, understanding, welfare

## Abstract

**Simple Summary:**

There are around 11 million pet dogs in the United Kingdom, yet there is more to learn about how best to ensure their welfare needs are met. We know that owners have an impact on their dog’s well-being and studies have suggested that what people do before they get their dog can subsequently impact their dog’s welfare. We used data collected by the People’s Dispensary for Sick Animals (PDSA) for the 2017 PDSA Animal Wellbeing (PAW) Report, the primary nationally representative means of monitoring the well-being of pets in the UK. We investigated associations between how much research people did before they got their dog and what they did or thought about. We found that owners who did more research before getting their dog were likely to be younger or have a higher education level. They were also more likely to have heard of the ‘five welfare needs’, have a realistic idea of the cost of owning a dog, and know the best place to go for help with their dog’s behaviour. These findings help us understand what influences dog owners and suggest that doing more research before getting a dog is linked to improved knowledge, understanding, and behaviours.

**Abstract:**

Despite many dogs living in homes in the UK, there is still more to know about the welfare of those individual animals. Past research has shown that owners’ thoughts and behaviours have a substantial impact on their dog’s welfare. This study aimed to better understand owners’ pre-purchase behaviours, knowledge and understanding, and ownership practices, and explore any associations between these factors and their dog’s welfare. We conducted further analysis of the data collected by People’s Dispensary for Sick Animals (PDSA) for their 2017 PDSA Animal Welfare (PAW) Report (*n* = 1814). We extracted variables to describe owner demographics (7), pre-purchase behaviours (1), knowledge and understanding (4), welfare indicators, (2) and ownership practices (4), and we tested for an association between these variables. We found more pre-purchase research was carried out by younger respondents and those with a higher education level. Also, more research was associated with feeling more informed about the five welfare needs and knowing to seek help for behaviour from appropriate sources. Overall, the study found several novel and significant results worthy of note and further exploration but did not find any strong connections between the variables.

## 1. Introduction

In the United Kingdom (UK), there are currently 11 million pet dogs, with 29% of adults owning a dog [1]. The Animal Welfare Act [2] introduced a duty of care for all owners to meet their pet’s five welfare needs, and the PAW Report was launched in 2011 as the first overarching means of identifying, assessing, and monitoring the well-being of companion animals and how their needs are being met. Given that owners are key to maintaining high levels of welfare in their dogs, it is vitally important to establish what owners understand about their welfare and what behaviours owners demonstrate daily to maintain or enhance their dog’s well-being.

In England and Wales, Section 9 of The Animal Welfare Act [2] states that the owner must take reasonable steps to ensure that they meet the needs of their animal. These needs include a suitable environment, a suitable diet, to exhibit normal behaviour patterns, to be housed with or apart from other animals, and to be protected from pain, suffering, injury, and disease. These five welfare needs are reflected in animal welfare policies in many countries. In England and Wales, they are enforceable by law and alongside additional clauses, aim to protect animals from abuse, cruelty, poor treatment, and neglect by people. Significant volumes of guidance for dog owners, including the ‘Code of practice for the welfare of dogs’, [3] have been produced by various interested parties including charities, local councils, and the government to advise owners on how best to meet their dog’s needs. Despite this extensive availability of information, many owners fail to meet their obligations to their pet dogs [4].

We know that the ability of dog owners to meet their animal’s welfare needs varies considerably and that a spectrum of ownership styles and abilities exist (and that these can change over time). It follows that factors such as an owner’s health, lifestyle, environment, access to resources, previous experiences, knowledge, understanding, and behaviours may all have a significant impact on their dog’s welfare. Therefore, an understanding of whether there are groups of owners who show the same behaviours or beliefs (ownership styles), whether there are any associations between an owner’s knowledge and understanding, and ownership practices, or whether certain factors such as owner demographics or pre-purchase behaviours relate to future behaviours is vital to understanding what affects the welfare of pet dogs.

It is often assumed that pre-purchase behaviours may be associated with subsequent ownership behaviours that impact a dog’s welfare. There is little research on owners’ thoughts and behaviours prior to dog ownership, indeed what has been done has largely relied on owners recalling the events months or years later. This lacuna is likely due to the challenges of identifying and accessing people who are considering getting a dog. Research that has been conducted has largely concentrated on people acquiring dogs from rescue centres and identifying factors predicting subsequent relinquishment [5,6,7] or has focused on motivations for acquiring specific dogs, i.e., age, breed, pedigree, rescue, etc. [8,9,10,11,12,13,14]. However, surveys have consistently shown that a significant number of people are acquiring dogs without conducting research into the implications of dog ownership, for example, a survey of 2000 new puppy owners by CEVA Animal Health in 2013 showed that 40% did no research before purchasing their dog [15]. More recently, the 2022 PAW Report also showed that 20% of pet owners did no research before purchasing their animal [16]. Similarly, a survey commissioned by the RSPCA [17] found that 14% of new dog owners did no research before buying a puppy, whilst a Dogs Trust survey showed that only 54% of owners had looked for advice or information before purchasing their dog [18]. Although two descriptive studies conducted in the US suggested that owners’ lack of research before acquisition is not necessarily associated with a higher risk of surrendering their dog [19,20], the RSPCA report concluded a link between impulse buying and relinquishment [17]. To date, no known studies have been conducted specifically looking into whether pre-acquisition research is linked to owner behaviour and subsequently their dog’s welfare.

It is expected that an owner’s knowledge and understanding of their dog’s needs are critical to that dog’s welfare [21]. Measuring the level of an owner’s knowledge of the five welfare needs is one way to establish their level of understanding. The 2022 PAW Report showed that only 14% of pet (dog, cat, and rabbit) owners had heard of the five welfare needs before taking part in the survey [16]. Equally, underestimating the cost of dog ownership has been associated with increased rates of relinquishment [17] and the 2022 PAW Report shows that only 18% of owners looked into the cost of pet ownership before acquiring their pet and 40% agreed that keeping a pet was expensive [16]. Another way to ensure that an owner is well-informed is determining if they know where to find reliable sources of assistance or information if required and the 2022 PAW Report found that only 6% of owners took advice from a veterinary professional before acquiring their pet [16].

The concept of ‘responsible dog ownership’ has been promoted by The Kennel Club [22] and the British Veterinary Association [23] as well as most other animal charities and UK councils. This concept includes ownership practices such as microchipping, neutering, registering with a vet, regular vaccinations, and having pet insurance. The PDSA PAW Reports show that a significant number of dog owners report not undertaking these ownership practices [21]. Similarly, Australian surveys such as those by Kobelt et al. [24], Masters and McGreevy [25], Howell et al. [26], and Rohlf et al. [27] showed that even committed dog owners do not follow some responsible ownership practices. Dog welfare can be improved directly through good ownership practices, including dogs spending less time alone, an appropriate frequency and duration of walks, and basic training. However, as the Code of Practice (DEFRA) [3] stated, “There is no one perfect way to care for all dogs because every dog is different and every situation is different”; it can be difficult to establish what is in each dog’s best interests. Most UK animal charities recommend that dogs should not routinely be left alone for more than four hours per day but all state that this guidance is dependent on the individual dog. The frequency and duration of walks required will also be dependent on the individual needs, but exercise and time exploring the outdoor environment are beneficial for most dogs’ physical and mental health. Bennett and Rohlf [28] investigated links between problematic behaviours, demographic variables, involvement in dog training, and participation in other dog–human interactions. They found that engagement in training and shared activities were predictive of less problematic behaviours but not necessarily as a result of these activities, and that perceived dog friendliness was associated with engagement in training. They therefore concluded that strategies to increase participation in dog training activities and promote canine sociability can have significant benefits for both dogs and owners [28]. Whilst obtaining and measuring good welfare involves a complex interaction and combination of multiple factors, a dog’s level of obesity can be seen as an objective indicator of health. Obese dogs have a shorter lifespan, a lower quality of life, and are more likely to suffer from additional medical conditions such as diabetes and osteoarthritis [29]. A study by Yam et al. [30] showed only 48% of dog owners could correctly estimate their dog’s body weight. To assist with this, Body Condition Scores (BCSs) have been developed but with limited success [31]. A study by Eastland-Jones et al. [32] demonstrated that most owners incorrectly estimate their dog’s body condition both with and without a BCS, with underestimation being the most common and that owners of overweight dogs were more likely to misjudge their dog’s BCS. Therefore, being able to objectively monitor a dog’s weight and body condition is thought to be an important factor in improving dog welfare [33], and developing an understanding of what affects this ability would be a logical first step.

Upjohn [34] recently suggested that to help owners make better decisions about their dog’s care, we need to better understand owner behaviour. Large-scale surveys can provide some insight into current owners’ behaviours and their dog’s welfare. PDSA launched the first PDSA Animal Wellbeing (PAW) Report in 2011, and it has become the primary means of identifying, assessing, and monitoring the well-being of companion animals in the UK [35]. Each year, PDSA works with the market research company YouGov^®^ to survey nationally representative samples of pet dog, cat, and rabbit owners, providing insight into animal welfare issues, estimating pet population numbers, and understanding how people care for their pets [21]. In addition, they intermittently survey veterinary professionals to gather their thoughts and opinions about key welfare issues. Since 2011, PDSA has provided valuable information suggesting that millions of UK pet owners are not appropriately meeting their animals’ welfare needs. The reports have predominantly described the percentages of responses and statistically significant differences between variables such as the year of survey and species of pet. More recently, further investigations into the associations between responses to questions have been explored [36,37], and this paper builds on this work. 

### 1.1. Aims 

Using the data from the 2017 PAW Report, this research aims to explore key dog ownership variables and identify groups of owners for whom human behaviour change interventions could be targeted to potentially inform and improve dog ownership practices and the welfare of pet dogs.

### 1.2. Research Questions

To achieve our aims, the following research questions were developed:To what extent do pre-purchase behaviours vary with owner demographics?Are differences in pre-purchase behaviours associated with differences in owner knowledge and understanding?To what extent do pre-purchase behaviours correlate with later ownership practices?Are variations in pre-purchase behaviours by owners associated with differences in welfare indicators in their dogs?Are differences in owner knowledge and understanding associated with differences in welfare indicators?Are specific ownership practices associated with differences in welfare indicators?

## 2. Methods

### 2.1. Ethical Approval 

Ethical approval for this research was obtained from the University of Bristol Faculty of Health Science Research Ethics Committee (Reference number 62861).

### 2.2. Population and Procedures

Anonymised data obtained from dog owners as part of the cross-sectional study of UK pet owners for the 2017 PAW Report [38] were analysed. The PAW Reports gather data through an anonymous online questionnaire. Surveys are conducted by YouGov^®^ using a nationally representative sample of UK adults (aged 18 and above) who own pets. YouGov^®^ recruit participants from various sources, such as advertising and partnerships with other websites. The survey panel comprises over 1 million people residing in the United Kingdom. Once recruited, participants are not obligated to participate in YouGov^®^ surveys and can only participate in any survey once. Participants receive a small incentive for participating, which is standard for online surveys. YouGov^®^ randomly select a sample of current pet owners from the online survey panel. Only those who own at least one pet dog, cat, or rabbit are eligible to participate. YouGov^®^ then sends out an email invitation to these individuals, along with a link to participate. Further information regarding data collection by YouGov^®^ for all PAW Reports including design, sampling, and measures can be found in Wensley, Betton, Gosschalk, Hooker, Main, Martin, and Tipton [36].

Data collection for the 2017 report was undertaken between 23 February and 6 March 2017. The survey was conducted online. Respondents with more than one pet were asked to complete the survey by selecting one at random and answering the survey based solely on that pet. Overall, there were 78 questions in total, but many of these were species (dog, cat, rabbit) specific so not all questions would have been asked of every participant. The survey took a maximum of 16 min to complete. The total sample size was 4153 dog, cat, and rabbit owners aged 18 and over who lived in the UK and, of those, 1814 were dog owners. Not every question received responses from all participants; therefore, the *n* value for each question has been presented where necessary.

### 2.3. Data Handling

Anonymised and coded raw data were received from PDSA in SPSS version 25. 

### 2.4. Extracted Variables

From the data, we extracted 18 key variables describing owner demographics (Table 1), pre-purchase behaviours (Table 2), knowledge and understanding (Table 3), welfare indicators (Table 4), and ownership practices (Table 5) to answer our research questions. The social grade of participants was calculated by YouGov^®^ based on the chief income earner and according to the social classification system used by the advertising industry and market research agencies [39]. We hypothesised an association between employment status, income, and social grade and found a correlation between the employment status of the respondent and household income (*n* = 1377, *r_s_* = −0.38, *p* < 0.0001), the employment status of the respondent and social grade (*n* = 1814, *r_s_* = 0.10, *p* < 0.0001), and between income and social grade (*n* = 1377, *r_s_* = −0.27, *p* < 0.0001). Although statistically significant, the correlation coefficients were very small, so we continued to test each of these variables separately.

The variable ‘number of sources of pre-purchase advice’ was devised from counting the number of sources of advice (Table 2) excluding both the ‘Nothing—I didn’t do anything’ and ‘Nothing—he/she was a present’ responses. 

We measured owner knowledge and understanding by respondents having a realistic estimation of the cost of dog ownership, how informed respondents felt about the five welfare needs, saying they would seek help from appropriate sources for problem behaviours and objectively checking their dog’s weight (Table 3). The variable entitled ‘realistic estimation of the lifetime cost of dog ownership’ was calculated from respondents’ answers to how much they thought their pet cost over their lifetime. Whilst it is acknowledged that this figure will vary depending on the dog’s size, etc., PDSA had estimated that, in 2017, the minimum cost of owning a small dog over their lifetime was £6500 [39]; therefore, responses below this figure were classified as an ‘unrealistic’ estimation and those over were classified as ‘realistic’. For the variable ‘level informed of the five welfare needs’, original answer options were ‘very well informed’, ‘quite well informed’, ‘not very well informed’, and ‘not at all well informed’. These responses were then categorised by YouGov^®^ to see what proportion felt informed about all of the five welfare needs. For the variable ‘seeks appropriate initial behavioural advice’, we established, through author consensus, that ‘appropriate responses’ to the answer options were to seek initial advice from a veterinary practice or behaviourist and a new variable was created showing those who reported they would seek advice or would not seek advice from an ‘appropriate source’. For the variable ‘objectively deciding dog’s weight’, we established, again through author consensus, that ‘objective responses’ to the answer options were for respondents to weigh their dogs or seek advice from a vet or vet nurse, and a new variable was created showing those who reported they would ‘objectively’ determine their dog’s weight.

Welfare indictors were the reported number of problem behaviours and the dog’s body shape (Table 4). The variable entitled ‘number of problem behaviours’ was calculated from the total number of behaviours their dog displays that the respondent would like to change. For the variable ‘combined dog’s body shape’ we devised the variable categorising ‘perfect shape’ responses compared to all others (very fat, fat, thin, and very thin). For this question, respondents were asked to look at pictures of dogs to help them to ‘describe’ their dog’s body shape (Figure 1).

Ownership practices were the daily time the dog was left alone, weekly walking time, attending formalised training, and the number of reported responsible ownership practices (Table 5). For the variable ‘daily time alone’ the original options were in hours per day and no additional calculations were required. The variable ‘weekly walking time’ was calculated from walking frequency and walking time to give a weekly walking time in minutes. As answer options for walking time were on a scale (for example, 11 to 30 min or 31 min to one hour), the upper limit of the walking time was used for the calculation (30 min and one hour). For the variable ‘attended formalised training’, we established that ‘formalised training’ included attending training classes or lessons with an expert and distinguished this from all other responses. The variable entitled ‘responsible ownership practices’ was calculated from the total number of positive responses to eight healthcare and responsible ownership practices presented. 

### 2.5. Statistical Analysis

Descriptive statistics are presented using frequencies and percentages, medians and interquartile ranges, or means and standard deviations (±), as appropriate. For any variables that included ‘don’t know’ or ‘not sure’ options, these responses were removed before analysis. 

We tested the association between pre-purchase behaviours and other key variables. To test our research questions (I-IV), we compared the frequency of pre-purchase behaviours with owner demographics (I), knowledge and understanding (II), ownership practices (III), and, finally, welfare indicators (VI). We also hypothesised and tested associations between owner knowledge and understanding, their subsequent ownership practices (behaviours) (V), and their dog’s welfare indicators (VI).

Most data collected were at nominal or ordinal level; therefore, non-parametric tests were generally employed. A parametric test (Pearson’s) was used to test associations involving seven variables. They were age and pre-purchase advice, pre-purchase advice and walking time, problem behaviours and time alone, problem behaviours and ownership practices, and, finally, problem behaviours and walk time. Between-group differences were tested using Chi-Squared (χ^2^), Mann–Whitney U (U), and Kruskal–Wallis (KWH). Correlations were tested using Spearman’s rho (*r_s_*) or Pearson’s (*r*). In comparisons, where median and quartiles were identical, means and standard deviations were reported to show differences. The *p*-value was set at a 0.05 level of significance.

To explore the existence of ownership styles, Principal Components Analysis (PCA) was undertaken using the continuous and ordinal variables: frequency of pre-purchase behaviour, the estimated lifetime cost of ownership, level of informed of the five welfare needs, number of healthcare and responsible ownership practices, number of problem behaviours, time left alone, walking frequency, and walking time. Owner demographics of age, education, and household income were also included. All other variables were nominal or ordinal and were not appropriate for this analysis. The correlation matrix was visually inspected to ensure each variable had a strong enough correlation to be included in further analysis; however, none of the variables reached the *r* ≥ 0.3 cut-off criteria. Similarly, the Kaiser–Meyer–Olkin (KMO) measure of sampling adequacy was 0.58, below the 0.6 minimum requirement. Both findings suggested that PCA was not appropriate for this data set.

## 3. Results

### 3.1. Descriptive Statistics

In total, 1814 responses were received, with 54.6% of the respondents being female. Most of the respondents (94.3%) identified as being White British and the average age was 54.22 (±14.9) years. The respondents were mainly working full-time (32.4%) or retired (34.4%). Overall, 61.3% were identified as being in a higher social grade (ABC1) and 38.7% as a lower social grade (C2DE). The median education level was reported as ‘qualifications below degree level’ and the median household income was £25,000 to £29,999 per year. The mean number of dogs owned was 1.37 (SD 1.04); most dogs were male (53.9%) and pedigree breeds (62.1%) (Table 6).

#### 3.1.1. Pre-Purchase Behaviours

Overall, 81.9% of the respondents reported seeking advice from at least one of the listed sources before they chose their dog (*n* = 1814, Table 7). The mean number of sources of information sought was 1.56 (±1.29).

#### 3.1.2. Knowledge and Understanding

Only 30.8% of the respondents reported a realistic estimation of the lifetime cost of dog ownership; the range of responses varied from £1 to £50,000,000 with a median of £4000 (£1500, £8000). Most reported feeling very informed (35%) or informed (47.4%) about the five welfare needs with 17.7% reporting not feeling informed (*n* = 1814). Only 37.2% reported they would seek initial behavioural advice from an appropriate source such as a vet or behaviourist, and 10.8% reported they would not seek advice from anywhere (*n* = 1162) (Table 5). Most (71.9%) reported that they decide if their dog is the correct weight objectively, by either weighing him/her or taking a vet or vet nurse’s advice while others mainly relied on less objective measures including common sense (33%), looking at their dog’s body (36.8%), or feeling their dog’s body (24.7%) (Table 8).

#### 3.1.3. Ownership Practices

The average time that respondents reported leaving their dog alone was 2.28 (±3.28) hours per day. The mean weekly walk time was calculated as 9.97 (±7.41) hours per week or 85.46 min per day. The respondents reported an average of 6.40 (±1.60) responsible ownership practices with 5.8% of owners reporting their dog was not microchipped, 7.1% reporting their dog was not registered with a vet, and 45.1% reporting their dog was not insured (Table 9). Only 30.7% of the respondents reported attending formal training with their dog and 11.2% reported that they had not trained their dog in any way (Table 10).

#### 3.1.4. Welfare Indicators 

Overall, 37.1% of the respondents described their dog as having the perfect body shape with only 17.8% reporting their dog to be fat or very fat and 45.1% describing their dog as thin or very thin. On average, respondents reported 1.11 (±1.33) problem behaviours with jumping up at people being the most frequently reported (22.3%) (Table 11). 

### 3.2. Association Testing

#### 3.2.1. To What Extent Do Pre-Purchase Behaviours Vary with Owner Demographics? 

There were no significant differences in the number of sources of pre-purchase advice with the gender of the respondent (*n* = 1814, U = 427,861, *p* = 0.06) or the region in which they lived (χ^2^(11) = 14.20, *p* = 0.22). There were, however, significant differences with employment status (KWH (7) = 18.51, *p* = 0.01); full-time students reported using the greatest number of sources (mean = 2.06 ± 1.30) and those who were unemployed reported the least (mean = 1.38 ± 1.42). There was a significant difference between the social grades ABC1 and C2DE, with ABC1 (higher social grade) reporting more sources of advice (*n* = 1814, U = 333,910.5, *p* < 0.0001, mean = 1.68 (±1.30) vs. mean = 1.37 (±1.23)). There was also a significant small positive correlation between the education level of the respondent and the number of sources of pre-purchase advice (*n* = 1762, *r*_s_ = 0.17, *p* < 0.0001), the higher the respondent’s education level, the more pre-purchase resources they reported using. There was a significant, yet small, positive correlation between household income and the number of sources of pre-purchase advice (*n* = 1811, *r*_s_ = 0.09, *p* < 0.0001). There was also a small significant negative correlation between the age of the respondent and the number of sources of pre-purchase advice (*n* = 1814, *r* = −0.11, *p* < 0.0001), with older respondents generally consulting fewer. 

#### 3.2.2. Are Differences in Pre-Purchase Behaviours Associated with Differences in Owner Knowledge and Understanding?

There was a significant difference between those who were realistic and unrealistic in their estimation of the cost of dog ownership in terms of the number of sources of pre-purchase advice consulted. Unexpectedly, those who were unrealistic reported to have consulted marginally more sources than those who were classified as realistic (*n* = 1814, U = 403,096.5, *p* < 0.0001, 1 (1, 2) vs. 1 (1, 3)). There was a small positive correlation between the number of sources of advice and the level of feeling informed about the five welfare needs (*n* = 1814, *r*_s_ = 0.05, *p* = 0.042), with more sources being associated with feeling more informed. Those who sought advice from a vet or behaviourist had generally consulted more sources (*n* = 1162, U = 1,708,186, *p* < 0.0001, 2 (1, 3) vs. (1 (1, 2)) than those who had not. There was a significant difference in the number of sources of advice between those who reported they objectively decided whether their dog was the correct weight and those who did not; those who objectively decided had, on average, sought pre-purchase information from more sources (*n* = 1814, U = 407,874.5, *p* < 0.0001, 1 (1, 2) vs. 1 (0, 2)).

#### 3.2.3. To What Extent Do Pre-Purchase Behaviours Correlate with Later Ownership Practices?

The correlation between the number of sources of pre-purchase advice and the number of hours respondents reported their dog was left alone was not significant (*n* = 1814, *r*_s_ = 0.05, *p* = 0.05). However, there was a small positive correlation between the number of sources of pre-purchase advice and the time respondents reported spending walking their dog per week (*n* = 1745, *r* = 0.08, *p* = 0.002); the more pre-purchase advice sources, the more time spent walking their dog. There was a significant difference in the number of sources of pre-purchase advice between those who reported they had undertaken formalised in-person dog training and those who had not. Those who had undertaken formalised in-person dog training had generally consulted more sources of pre-purchase advice (*n* = 1814, U = 437,066, *p* < 0.0001, 1 (1, 3) vs. 2 (1, 2)). There was also a small positive correlation between the number of sources of pre-purchase advice and the number of responsible dog ownership practices reported (*n* = 1814, *r*_s_ = 0.15, *p* < 0.0001). 

#### 3.2.4. Are Variations in Pre-Purchase Behaviours by Owners Associated with Differences in Welfare Indicators in Their Dogs?

There was a small positive correlation between the number of sources of pre-purchase advice and the number of problem behaviours reported (*n* = 1814, *r*_s_ = 0.11, *p* < 0.0001). There was no association between the number of sources of pre-purchase advice and whether the respondent reported their dog as having a perfect body shape or not (*n* = 1814, U = 389,919.5, *p* = 0.56).

#### 3.2.5. Are Differences in Owner Knowledge and Understanding Associated with Differences in Welfare Indicators?

There was no significant difference in the number of problem behaviours reported (*n* = 1814, U = 363,878.5, *p* = 0.18) or the likelihood of reporting a correct body shape (*n* = 1814, χ^2^(1) = 0.004, *p* = 0.949) between those who had a realistic estimation of the cost of dog ownership and those that did not. There was a small negative correlation between the number of problem behaviours reported and how informed respondents were about the five welfare needs (*n* = 1814, *r*_s_ = −0.18, *p* < 0.0001); in general, those who were less informed reported more problem behaviours. Those who reported their dog had a perfect body shape showed no difference in how informed they felt about the five welfare needs compared to the rest of the population (*n* = 1814, χ^2^(2) = 4.83, *p* = 0.09). There was no difference in the reported number of problem behaviours between those who said they would seek behavioural advice from a veterinary practice or behaviourist and those who did not (*n* = 1162, U = 166,333.5, *p* = 0.09). 

#### 3.2.6. Are Specific Ownership Practices Associated with Differences in Welfare Indicators?

There was a small positive correlation between the reported number of problem behaviours and the time the dog was left alone (*n* = 1814, *r* = 0.08, *p* = 0.001); in general, the more time left alone, the more problem behaviours. There was a small negative correlation between the reported walking time and the number of problem behaviours reported (*n* = 1745, *r* = −0.09, *p* < 0.0001); the more walking time reported, the fewer problem behaviours. Contrary to expectations, those who reported they had undertaken formalised dog training reported a higher number of problem behaviours than those who had not (*n* = 1814, U = 392,041.5, *p* < 0.0001, 1 (0, 2) vs. 1 (0, 2), mean = 1.32 (±1.46) vs. 1.02 (±1.26)). There was no correlation between the number of responsible ownership practices and the number of problem behaviours reported (*n* = 1814, *r* = −0.02, *p* = 0.31).

## 4. Discussion

### 4.1. Summary of Findings

To achieve our aims of exploring dog ownership variables and answer our research questions, we received a total of 1814 anonymised dog owner responses from the PDSA 2017 PAW Report [38] data set that comprised data collected from a nationally representative sample of UK pet-owning adults. We identified and investigated the key dog ownership variables provided by the survey. When testing for associations between pre-purchase behaviour and other variables, we found five significant associations with owner demographics, four with owner knowledge and understanding, three with ownership practices, and one with welfare indicators. We also found one significant association between the variables that tested owner knowledge and understanding and welfare indicators and three significant associations between ownership practices and welfare indicators. Overall, the study found several novel and significant associations worthy of note and further exploration but did not find any strong connections between the variables that were examined.

#### 4.1.1. Pre-Purchase Behaviours

More than one in ten (11.5%) dog owners reported not seeking advice from any sources before acquiring their dog and 3.6% reported that their dog was a present. This situation is reflected in the findings of all pet owners (dog, cat, and rabbit) in the PAW Reports over the years [16,21,38,40] as well as surveys by CEVA Animal Health [15], the RSPCA [17], and the Dogs Trust [18]. This proportion was, however, considerably lower than found by Kuhl et al. [41], who reported 21.1% of 1066 UK owners had not sourced information about their dog(s) before purchasing them. It is also lower than reported by Dogs Trust 2019, whose online survey of current (*n* = 8050) and potential (*n* = 2884) dog owners, showed that only 54.4% of the current dog owners had looked for advice or information before acquiring their dogs [18]. This discrepancy in findings may reflect how truly variable this factor is, but may also be due to sampling bias, recall bias, differences in the phrasing of questions, how invested owners are in their dogs. or inaccuracy in the information reporting.

Only 6.2% of our respondents reported seeking advice from veterinary professionals, with 33.6% looking on the internet for advice. These findings are supported by a recent cross-sectional online survey that found that many respondents (18.8%) did a general internet search but only 3.8% contacted a veterinary practice before acquiring their dog (*n* = 895) [41]. Another online survey also found that most (78.6%, *n* = 571) owners reported using the internet for pet health information rather than a vet [42]. Given the significant potential for obtaining inaccurate information online, these findings suggest a concerning trend and more research is required to establish whether those who seek information online are doing so from reliable sources. Equally, having established a pre-acquisition relationship with a vet will likely provide improved and contextualised future care for the individual dog.

Out of the respondents, 41.5% claimed to have previous experience with the breed or owning a dog. It is important to note that this experience may not necessarily be relevant when searching for a new dog, since each is unique. These findings regarding previous experience are, however, supported by a recent large-scale online survey whose multivariable analysis showed that previous dog ownership was an important factor in determining the likelihood of pre-acquisition research being undertaken [18]. It showed that those with previous experience of dog ownership were the least likely to undertake research (44.8% of current owners and 77.4% of potential owners). This study also compared those who had owned a dog as an adult and as a child and showed that those who lived with a dog as an adult were 1.2 times more likely amongst current owners and 1.4 times more likely amongst prospective owners to conduct research. Those who lived with a dog as a child were 2.5 times (current) and 3.6 times (prospective) more likely to conduct research. These findings all suggest that pre-purchase behaviours may provide some valuable insight into dog acquisition and future ownership practices and that further research into this area would be of interest. 

#### 4.1.2. To What Extent Do Pre-Purchase Behaviours Vary with Owner Demographics? 

Apart from the respondent’s gender and the region of the UK in which they lived, all other owner demographics showed statistically significant associations with the amount of pre-purchase advice sought. Generally, the older the respondent, the less pre-purchase research they did. This is probably because the older respondents were more likely to have previous dog ownership experience and therefore less likely to feel the need to seek advice before purchasing their dog; however, this hypothesis was not tested as part of this study. Equally, as most information is now available online, this may be inaccessible to the older generation. These findings are supported by the study by the Dogs Trust which established through multivariable analysis that age was a significant factor in whether research was undertaken, with younger prospective dog owners more likely to undertake research [18]. Just one -third of current owners aged 75 or older undertook any research prior to acquiring their most recent dog compared to 67.2% of 25–34-year-olds and 65.6% of 18–24-year-olds. They also reported that owners aged 25–34 were 2.3 times more likely to have undertaken research compared to those aged 75 years or older.

Full-time students reported the greatest number of sources and those who were unemployed reported the least. There was a difference between the social grades, with those in higher social grades (ABC1) reporting more sources compared to lower social grades (C2DE). Those with a higher education level or higher household income also tended to report consulting more sources of pre-purchase advice. As reported in Section 2.5, a significant yet very small positive correlation was found between the variables of household income, employment status, social grade, and education level, which may explain the similarities in these findings. It is presumed that full-time students are likely to be working towards higher education levels and that in general, students are likely to be younger and have less experience in dog ownership, but are more likely to have the knowledge and skills to search for information. Again, some of these findings are supported by Mead et al. [18], who found that respondents with formal education levels were significantly more likely to have undertaken research before getting their dog. They found that only one third of those who had no formal qualifications had undertaken research. They calculated that the odds of undertaking research increased with the level of education and that those with a postgraduate qualification had 2.3 times greater odds of having undertaken research (63.3%). We can therefore conclude that pre-purchase behaviours can be associated with some owner demographic profiles and that this is a question worthy of ongoing investigation as well as providing useful information as to whom animal welfare charities may wish to target in future campaigns.

#### 4.1.3. Are Differences in Pre-Purchase Behaviours Associated with Differences in Owner Knowledge and Understanding?

Those who had sought a greater number of sources of pre-purchase advice were likely to feel more informed about the five welfare needs; report that they would seek help for behavioural problems from a vet or behaviourist, and be more likely to use an objective measure to decide whether their dog is the correct weight. Inexplicably, however, they were more likely to have an unrealistic estimation of the cost of dog ownership. Therefore, overall, we can conclude an association between pre-purchase advice-seeking behaviours and subsequent levels of knowledge and understanding. Whilst no known studies have looked directly at these specific associations, it follows that individuals who actively seek more information pre-purchase are likely to continue to seek out information post-acquisition. A study of UK dog owners noted changes in information sourcing from pre-acquisition to information accessed during the dog’s lifetime and showed that dog owners accessed a large variety of often non-verified sources of information [41]. It should also be noted that seeking advice and changing behaviour as a result of that advice involves a complex interaction of multiple variables [43]. Finding information at any time point can lead to increased levels of knowledge and understanding of dog ownership but, as Kuhl et al. [41] state, “While the sourcing of pre-purchase advice contributes to owners making evidence-based, welfare friendly decisions, reputable rather than anecdotal information sourcing needs to continue throughout a dogs’ lifetime, to have a positive influence”.

A survey commissioned by the RSPCA highlighted that underestimating the cost of dog ownership may be one factor that can lead to relinquishment and concluded that pre-purchase behaviours may play a part in this [17]. The 2022 PAW Report showed that only 18% of pet owners looked into the cost of pet ownership before acquiring their pet and 40% thought that keeping a pet was expensive and only 14% of pet owners had heard of the five welfare needs [16]. Studies have also shown that owners who demonstrate less knowledge and understanding are less likely to seek help from reliable sources compared to those with more knowledge and understanding [41,42], and that there are significant challenges in deciding whether or not their dog is the correct weight [30,32]. This study confirms their assertions and shows an association between these factors and pre-acquisition behaviours, something that has only previously been hypothesised. 

#### 4.1.4. To What Extent Do Pre-Purchase Behaviours Correlate with Later Ownership Practices?

Those who had sought more sources of pre-purchase advice were more likely to report longer weekly walking times and to undertake formalised in-person dog training. Therefore, overall, we can conclude an association between more extensive pre-purchase research and positive later ownership practices. We know of no previous studies exploring these associations. Our findings suggest that pre-purchase behaviours are associated with some subsequent ownership behaviours, which again implies that potential owners might be better prepared for the practicalities of dog ownership by carrying out more pre-acquisition research or that they are the type of individual to continue to seek out information or support if required. These findings show an interesting association between factors that had previously been assumed [17] and an association worthy of ongoing exploration in future studies. 

#### 4.1.5. Are Variations in Pre-Purchase Behaviours by Owners Associated with Differences in Welfare Indicators in Their Dogs?

Pre-purchase behaviours had no association with whether the respondents reported their dog as having the perfect body shape. However, the reliability of owners accurately estimating their dog’s body weight or body condition score is known to be low [30,32]; therefore, these findings may not be a true reflection of any (lack of) association. Interestingly, however, 37.1% of the respondents reported their dog to be the perfect body shape, which is similar to findings in previous studies [44]. In contrast, those who had sought more sources of pre-purchase advice reported more problem behaviours in their dog. This may be a result of increased awareness of potential behavioural problems or reflective of an increase in knowledge obtained through research more generally, or that they are the type of individual to continue to seek out support if required. However, it may have been predicted that those who had sought more sources of pre-purchase advice were better prepared for dog ownership and more able to prevent problem behaviours from occurring, contradicting these current findings. Overall, this question has raised some unexpected and interesting findings, but further exploration of this issue is required. 

#### 4.1.6. Are Differences in Owner Knowledge and Understanding Associated with Differences in Welfare Indicators?

No association was found between whether the respondents reported their dog being the perfect body shape (or not) and having a realistic estimation of the cost of dog ownership, how informed they were about the five welfare needs, whether they would appropriately seek help for problem behaviours, nor whether they objectively measure their dog’s weight. The lack of association between objectively measuring their dog’s body weight and reporting the perfect body shape may be a result of the limitations in owners correctly estimating their dog’s body weight or body condition score [30,32]. Only 17.8% of the respondents reported their dog was fat or very fat, whereas 45.1% described their dog as thin or very thin, given the current obesity epidemic, these reports seem unlikely to be accurate [33]. Similarly, no association was found between the number of problem behaviours reported and having a realistic estimation of the cost of dog ownership, appropriately seeking help for problem behaviours, and objectively checking their dog’s weight. However, those who reported feeling informed about the five welfare needs were likely to report fewer problem behaviours. This single association supports the hypothesis that the more knowledge of their dog’s needs an owner has, the more able they are to prevent or correct problem behaviours; however, this issue requires ongoing study and further testing to conclude whether this is just an association or whether cause and effect can be demonstrated.

#### 4.1.7. Are Specific Ownership Practices Associated with Differences in Welfare Indicators?

No association was found between whether the respondent reported their dog being the perfect body shape and the daily time alone, weekly walking time, attending formalised training, or the number of responsible ownership practices. There was no significant correlation between the number of responsible ownership practices and the number of problem behaviours. However, those who reported a greater weekly walking time reported fewer problem behaviours and those who reported more time alone reported more problem behaviours. The correlation between time alone and problem behaviours may be due to many of the behaviours listed in the survey being associated with separation-related behaviours (SRBs). SRBs are unwanted dog behaviours that only occur in the perceived absence of their owners [45]. The behaviours often include vocalisation, destructive behaviour, and inappropriate elimination (urination and defaecation) [46,47], all of which were listed. These are more likely to be experienced if dogs are left alone more often. Equally, a lack of companionship and/or stimulation may also account for some of the reported problem behaviours, and the alternative association between greater walking time and fewer problem behaviours may also support this theory. However, these are currently only reported associations and a causative link cannot be assumed without further investigation.

Interestingly, those who reported attending formalised training were likely to report more problem behaviours. These findings contradict those of Bennett and Rohlf [28], who showed that engagement with training activities was predictive of fewer problematic behaviours. However, it may be that owners who experienced more problematic behaviours were more likely to attend formalised training to manage those behaviours. Alternatively, it might reflect the lack of regulation of trainers, which means that some will not be providing evidence-based advice, may be using out-of-date techniques or positive punishment, or that training helps to raise awareness of problem behaviours. Overall, we can identify several interesting links between ownership practices and welfare indicators that warrant further exploration. 

### 4.2. Study Limitations

The accuracy of the survey responses to some questions may be affected by the individual completing the survey, e.g., if they are not the primary caregiver, although options for ‘Don’t know’ are made available, where applicable, to allow for this. When it comes to pre-purchase behaviour, questions heavily depend on memory. This can be challenging for owners who have had their dogs for several years and may need to recall information from a long time ago. This can lead to less reliable responses and suggests the need for real-time prospective studies. Further study could filter the sample to look only at those who had acquired their pet recently.

Some of the questions and answer options in the PDSA survey were phrased somewhat ambiguously, raising questions about the findings and limiting options for in-depth data analysis. For example, for the question regarding where someone would seek initial behavioural advice, they were told to ‘tick all that apply’. Whilst it can be difficult for respondents to pick just one option, it may have been preferable to construct the question in that way to ascertain what the first source of initial advice was. Similarly, the question relating to behaviours could be viewed as having two components (‘which behaviours does your dog display that you want to change’) combined into one question, whereas these could be perceived as two quite different concepts as well as being very subjective. Such questions have since been developed and improved in subsequent PAW Report surveys.

When creating new variables from existing responses, some consensus decisions were required to identify ‘good’ or ‘correct’ practices and these could be debated further. For example, when creating the variable ‘seeks appropriate initial behavioural advice’, we determined that ‘appropriate responses’ to the answer options were to seek initial advice from a veterinary practice or behaviourist but acknowledge that an appropriately qualified dog trainer or even a reputable source of online information may be ‘appropriate’. Also, for the variable ‘formalised dog training’, the answer options available may have been interpreted very differently by respondents and the content and quality of training would have varied considerably. Unfortunately, the data collected for these questions in 2017 did not explore the levels of training or quality of sources as it has done in subsequent years. Similarly, for the variable ‘objectively deciding the dog’s body weight’, we established that an ‘objective response’ to the answer options was that the respondents weighed their dogs or sought advice from a vet or vet nurse, but acknowledge that, with an understanding of BCSs, the look or feel of a dog’s body could also be considered an objective measure. It should also be noted that the images used to represent the dog’s body condition (Figure 1) do not represent the standard body condition images, and elements such as differences in the tail position between the pictures may have biased some respondents. Therefore, caution should be given to the interpretation of any findings. These images have since been updated by PDSA to reflect the current standards. For future studies, exploration of the ‘quality’ of the sources of information should be considered.

Much of the information collected and many of the variables analysed in this study resulted in binary data and were not optimal for many types of analysis. It should also be noted that whilst statistically significant associations were found between some variables, these were all based on very small correlations and were only likely to be significant due to the large sample sizes. Since this was a preliminary exploration, no correction factors were applied and as no strong associations were found, the results should be treated with caution. In reality, the relationships between the variables are likely complex, and hence, analyses exploring two variables at a time are likely to show strong relationships or fully predict owner behaviour and dog welfare. However, the relationships seen do suggest that these are questions were worthy of testing, and these findings should inform future multivariate studies. We also attempted a PCA with key variables to explore the existence of ownership styles, but the data set was found to be unsuitable for this analysis. However, as several significant associations were found, future studies should consider a PCA with revised questions (e.g., with more ordinal and continuous variables) where possible.

## 5. Conclusions

We utilised the dog owner data collected for the 2017 PAW Report and analysed those responses in greater depth. We explored key dog ownership variables and tested for the existence of dog ownership styles among groups. We found no strong associations between variables, and we were unable to identify any specific ownership styles, making it difficult to identify groups of owners for whom potential interventions could be targeted. However, weak associations were found between pre-purchase behaviours and knowledge and understanding as well as pre-purchase behaviours and later ownership practices. We also found some associations between ownership practices and welfare indicators, showing that these novel research questions are creditable and require further research. We would therefore recommend that future research focuses on the further investigation of associations between dog owners’ pre-purchase behaviours, ownership practices, and knowledge and understanding and that dog’s welfare.

## Figures and Tables

**Figure 1 animals-14-00396-f001:**
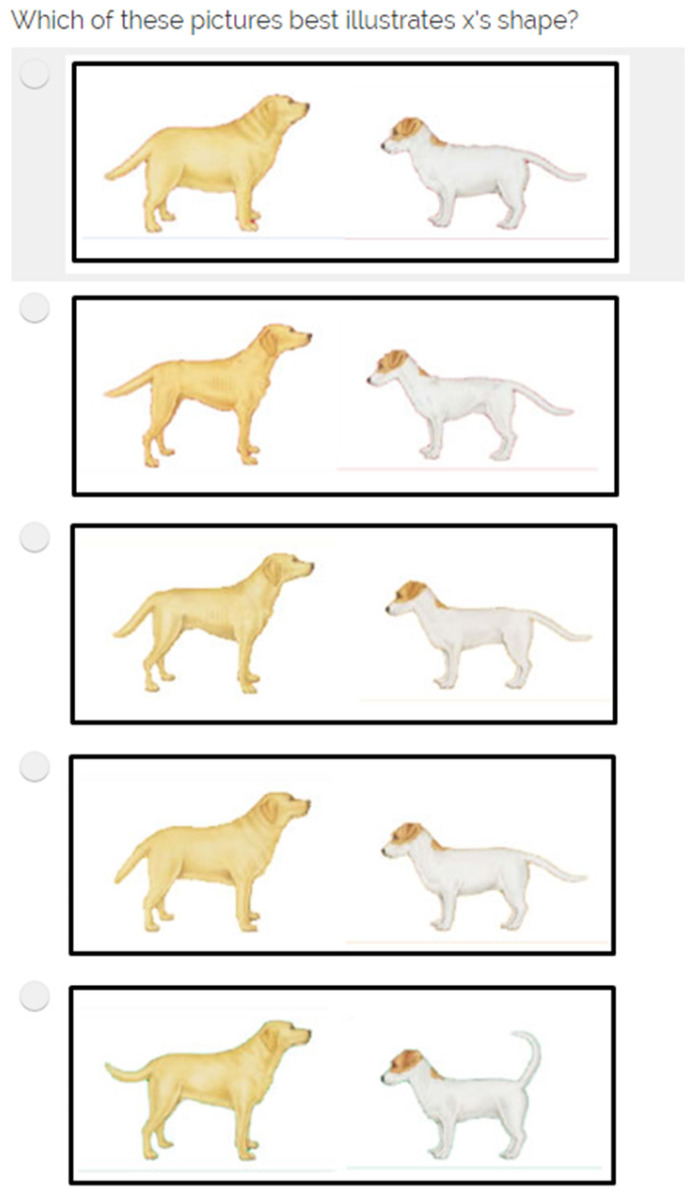
Pictures used to illustrate dog’s body condition including very fat, very thin, thin, fat, and perfect [39].

**Table 1 animals-14-00396-t001:** Respondent demographic details with survey questions, answer options, and coding.

Question	Answer	Variable Name	Coding	Level of Measurement
Which area of the UK do you live in?	North East	Region	1	Nominal
	North West		2	
	Yorkshire and the Humber		3	
	East Midlands		4	
	West Midlands		5	
	East of England		6	
	London		7	
	South East		8	
	South West		9	
	Wales		10	
	Scotland		11	
	Northern Ireland		12	
Are you male or female?	Male	Gender	1	Nominal
	Female		2	
How old are you?	Age in years	Age		Ratio
What is the highest educational or work-related qualification you have?	No formal qualifications		1	Ordinal
	Qualifications below degree level		2	
	University degree or higher		3	
Gross household income is the combined income of all those earners in a household from all sources, including wages, salaries, or rents and before tax deductions. What is your gross household income?	Under £5000 per year	Household income	1	Ordinal
	£5000 to £9999 per year		2	
	£10,000 to £14,999 per year		3	
	£15,000 to £19,999 per year		4	
	£20,000 to £24,999 per year		5	
	£25,000 to £29,999 per year		6	
	£30,000 to £34,999 per year		7	
	£35,000 to £39,999 per year		8	
	£40,000 to £44,999 per year		9	
	£45,000 to £49,999 per year		10	
	£50,000 to £59,999 per year		11	
	£60,000 to £69,999 per year		12	
	£70,000 to £99,999 per year		13	
	£100,000 to £149,999 per year		14	
	£150,000 and over		15	
	Don’t know		16	
	Prefer not to answer		17	
Social grade	ABC1 (higher grade)	Social grade	1	Nominal
	C2DE (lower grade)		2	
Which of these applies to you?	Working full time (30 or more hours per week)	Employment status	1	Nominal
	Working part time (8–29 h a week)		2	
	Working part time (Less than 8 h a week)		3	
	Full time student		4	
	Retired		5	
	Unemployed		6	
	Not working		7	
	Other		8	
How many people, including yourself, are there in your household? Please include both adults and children.	1	Household size	1	Ordinal
	2		2	
	3		3	
	4		4	
	5		5	
	6		6	
	7		7	
	8 or more		8	
	Don’t know		9	
	Prefer not to say		10	

**Table 2 animals-14-00396-t002:** Pre-purchase behaviour with survey questions, answer options, and coding.

Question	Answer	Variable Name	Coding	Level of Measurement
Which, if any, of the following did you do before you chose [dog’s name]? This might include deciding whether or not to get a pet, what type of pet to get, and where to get the pet from. Please tick all that apply.	Took advice from a veterinary professional e.g., vet, vet nurse or vet practice *	Pre-purchase source of advice	1/0	Nominal
	Took advice from friends or family *		1/0	
	Took advice from breeder *		1/0	
	Took advice from pet shop *		1/0	
	Took advice from rescue centre *		1/0	
	Looked on the internet *		1/0	
	Looked on social media *		1/0	
	Looked in book(s)/magazines/newspapers *		1/0	
	Had previous experience of the breed/animal *		1/0	
	Got advice from animal charity *		1/0	
	Other *		1/0	
	Nothing—he/she was a present *		1/0	
	Nothing—I didn’t do anything *		1/0	

* Answer options yes/no.

**Table 3 animals-14-00396-t003:** Knowledge and understanding with survey questions, answer options, and coding and calculated variables shown in italics.

Question	Answer	Variable Name	Coding	Level of Measurement
How much do you think your pet will cost you in total over his or her lifetime?	Answer to nearest pound	Lifetime cost of dog ownership		Ratio
	*Answered more than £6500*	*Realistic estimation of lifetime cost dog ownership*	1	*Nominal*
	*Answered less than £6500*	*Unrealistic estimation of lifetime cost dog ownership*	0	
How well informed do you feel about each of the areas below? That is, how to provide your pet with… A suitable place to live (environment). A proper diet (diet). The ability to express normal behaviour (behaviour). The need to be housed with or apart from other animals (company). Protection form and treatment of illness and injury (health and wellbeing).	Feel VERY informed about ALL of the five welfare needs *	Level informed of welfare needs	2	Ordinal
	Feel informed about ALL of the five welfare needs *		1	
	Not informed about ALL of the five welfare needs *		0	
If you were to seek help to change any of [dog’s name]’s behaviours, where would you go for initial advice? Please tick all that apply.	Online search engine e.g., Google *	Seek initial behavioural advice	1/0	Nominal
	Specific website *		1/0	
	A book *		1/0	
	A veterinary practice *		1/0	
	Friends and family *		1/0	
	A behaviourist *		1/0	
	A trainer *		1/0	
	Other *		1/0	
	None of these, I would not seek advice from anywhere *		1/0	
	Don’t know *		1/0	
	*Veterinary practice or behaviourist* *(Answered yes to veterinary practice or behaviourist)*	*Seeks appropriate initial behavioural advice*	*1/0*	*Nominal*
How do you decide whether or not your dog is the correct weight? Please tick all that apply.	Vet or veterinary nurse advice *	Deciding correct weight	1/0	Nominal
	Pet breeder advice *		1/0	
	Friends/relative advice *		1/0	
	Common sense *		1/0	
	Look at my pet’s body *		1/0	
	Feel of my pet’s body *		1/0	
	Weigh him/her *		1/0	
	Another way *		1/0	
	I’m never sure what the right weight is *		1/0	
	Not applicable—I don’t think about it *		1/0	
	*Deciding correct weight* *(Answered yes to vet or veterinary nurse advice and weigh him/her)*	*Objectively deciding correct weight*	1/0	*Nominal*

* Answer options yes/no.

**Table 4 animals-14-00396-t004:** Welfare indicators with survey questions, answer options, and coding, and calculated variables shown in italics.

Question	Answer	Variable Name	Coding	Level of Measurement
Which of the following behaviours, if any, does [dog’s name] display that you would like to change? Please tick all that apply.	Barking or vocalising for more than five minutes at a time when someone is present *	Problem behaviours	1/0	Nominal
	Growling or snarling *		1/0	
	Biting other dogs *		1/0	
	Jumping up at people *		1/0	
	Aggression towards people *		1/0	
	Aggression towards other pets *		1/0	
	Inappropriate toileting in the house *		1/0	
	Inappropriate sexual mounting (“humping”) *		1/0	
	Urinating when excited *		1/0	
	Destructive behaviour *		1/0	
	Signs of distress when left alone eg scratching, destructive behaviour, barking or howling for more than five minutes or toileting in the house *		1/0	
	Not coming back when called *		1/0	
	Chewing items not designed for chewing *		1/0	
	Showing signs of fear *		1/0	
	Other *		1/0	
	None of these *		1/0	
	*Number of problem behaviours calculated from problem behaviours (Excluding ‘none of these’)*	*Number of problem behaviours*		*Ratio*
Which of these pictures best illustrates {dog’s name]’s shape? (Figure 1)	Very thin	Dog’s body shape	1	Nominal
	Thin		2	
	Perfect		3	
	Fat		4	
	Very fat		5	
	*Perfect*	*Dog’s body shape perfect*	*1*	*Nominal*
	*Other* *(Calculated from very thin, thin, fat and very fat)*		*0*	

* Answer options yes/no.

**Table 5 animals-14-00396-t005:** Ownership practices with survey questions, answer options, and coding, and calculated variables shown in italics.

Question	Answer	Variable Name	Coding	Level of Measurement
Generally, how many hours is [dog’s name] left alone (with no human company) in the house during a weekday? Please do not include the time when you are asleep in bed. Please type in to the nearest hour. If you dog is never left, please type 0.	Time in hours	Time alone		Ratio
How often do you typically take [dog’s name] out for a walk?	More than once a day (14 times per week)	Walking frequency	8	Ordinal
	Once a day (7 times per week)		7	
	Every other day (5 times per week)		6	
	Four times a week		5	
	Three times a week		4	
	Twice a week		3	
	Once a week		2	
	Less often than once a week (0.5 times per week)		1	
	Never		0	
	Has free range (Excluded from analysis)		99	
When you walk [dog’s name], how long is this typically for?	Up to 10 min (10)	Walking time	1	Ordinal
	11–30 min (30)		2	
	31 min to 1 h (60)		3	
	1–2 h (120)		4	
	Over 2 h (180)		5	
	*Calculated from walking frequency x walking time (minutes)*	*Weekly walking time*		*Ratio*
In which of the following ways, if any, have you trained [dog’s name]? Please tick all that apply.	Completed a course through a regular dog training class *	Training	1/0	Nominal
	Went to one or more organised training class(es) *		1/0	
	Had a one off one to one lesson with an expert *		1/0	
	Had a course of one-to-one lessons with an expert *		1/0	
	Applied previous experience of how to train a dog *		1/0	
	Used an online or digital training programme *		1/0	
	Other *		1/0	
	None of these—my dog was already trained when I got him/her *		1/0	
	None of these—I haven’t trained my dog in anyway *		1/0	
	*Formalised dog training* *(Answered yes to training classes or lessons with expert)*	*Formalised dog training*	*1/0*	*Nominal*
Which, if any, of these has your pet had done? Has your pet been…	Neutered (spayed/castrated/snipped/done/dressed) *	Responsible ownership practices	1/0	Nominal
	Vaccinated—primary course (when young) *		1/0	
	Vaccinated—regular boosters/injections *		1/0	
	Microchipped *		1/0	
	Insured *		1/0	
	Wormed *		1/0	
	Treated for fleas *		1/0	
	Currently registered with a vet *		1/0	
	None of these *		1/0	
	*Number of healthcare and responsible ownership practices calculated from healthcare and responsible ownership practices (excluding ‘None of these’)*	*Total responsible ownership practices*		*Ratio*

* Answer options yes/no.

**Table 6 animals-14-00396-t006:** Frequency counts, percentages, medians (IQR), and means (SD) of the respondent (owner) demographics and dog details (*n* = 1814).

Question	Answer	%	Mean	SD	Median	IQR
Gender	Male	45.4				
	Female	54.6				
Age			54.22	14.9		
Ethnicity	White British	94.3				
	Any other white background	3.4				
	White and Black Caribbean	0.3				
	White and Black African	0.1				
	White and Asian	0.2				
	Any other mixed background	0.3				
	Indian	0.2				
	Any other Asian background	0.2				
	Black Caribbean	0.1				
	Black African	0.1				
	Any other black background	0.1				
	Chinese	0.2				
	Other ethnic group	0.3				
	Prefer not to say	0.5				
Religion	No religion	46.2				
	Christian	29.7				
	Other	23.9				
	Missing	0.3				
Region	North East	5.4				
	North West	11.2				
	Yorkshire and the Humber	10.1				
	East Midlands	9.6				
	West Midlands	7.7				
	East of England	9.3				
	London	5.0				
	South East	12.6				
	South West	12.5				
	Wales	3.8				
	Scotland	9.3				
	Northern Ireland	3.5				
Employment status	Working full time	32.4				
	Working part time	16.1				
	Retired	34.4				
	Unemployed or not working	11.5				
	Full time student	2.8				
	Other	2.9				
Social grade	ABC1 (higher social grade)	61.3				
	C2DE (lower social grade)	38.7				
Education level					2 or qualifications below degree level	1
Household size	*n* = 1797				2	1
Household income	*n* = 1377				6 or £25,000 to £29,999 per year	6
Number of dogs			1.37	1.04		
Sex of dog	Male	53.9				
	Female	46.0				
	Not sure	0.1				
Breed of dog	Cross breed (mixed/mongrel/hybrid)	27.4				
	Specialist cross breed (e.g., labradoodle)	7.9				
	Pedigree	62.1				
	Not sure	2.6				

**Table 7 animals-14-00396-t007:** Reported sources of information sought before purchasing their dog (*n* = 1814).

Answer	Frequency	%
Took advice from a veterinary professional e.g., vet, vet nurse or vet practice	112	6.2
Took advice from friends or family	325	17.9
Took advice from breeder	288	15.9
Took advice from pet shop	22	1.2
Took advice from rescue centre	278	15.3
Looked on the internet	609	33.6
Looked on social media	115	6.3
Looked in book(s)/magazines/newspapers	226	12.5
Had previous experience of the breed/animal	752	41.5
Got advice from animal charity	106	5.8
Other	115	6.3
Nothing—he/she was a present	66	3.6
Nothing—I didn’t do anything	208	11.5

**Table 8 animals-14-00396-t008:** Reported sources of initial behavioural advice (*n* = 1162) and method of deciding whether their dog is the correct weight (*n* = 1814).

Variable	Answer	% Yes	% No
Initial sources of behavioural advice	Online search engine e.g., Google	35.0	65.0
	Specific website	12.1	87.9
	A book	13.7	86.3
	A veterinary practice	24.4	75.6
	Friends and family	14.6	85.4
	A behaviourist	16.8	83.2
	A trainer	27.0	73.0
	Other	2.5	97.5
	None of these, I would not seek advice from anywhere	10.8	89.2
	Don’t know	4.6	95.4
Methods of deciding whether their dog is the correct weight	Vet or veterinary nurse advice	54.1	45.9
	Pet breeder advice	2.3	97.7
	Friends/relative advice	1.6	98.4
	Common sense	33.0	67.0
	Look at my pet’s body	36.8	63.2
	Feel of my pet’s body	24.7	75.3
	Weigh him/her	37.1	62.9
	Another way	1.3	98.7
	I’m never sure what the right weight is	0.3	99.7
	Not applicable—I don’t think about it	1.5	98.5

**Table 9 animals-14-00396-t009:** Responsible ownership practices reported (*n* = 1814).

Answer	Frequency	%
Neutered	1323	72.9
Vaccinated—primary course (when young)	1341	73.9
Vaccinated—regular boosters/injections	1462	80.6
Microchipped	1708	94.2
Insured	996	54.9
Wormed	1607	88.6
Treated for fleas	1489	82.1
Currently registered with a vet	1685	92.9
None of these	4	0.2

**Table 10 animals-14-00396-t010:** Different ways of training reported by respondents (*n* = 1814).

Answer	% Yes	% No
Completed a course through a regular dog training class	13.3	86.7
Went to one or more organised training class(es)	17.6	82.4
Had a one off one-to-one lesson with an expert	5.3	94.7
Had a course of one-to-one lessons with an expert	5.4	94.6
Applied previous experience of how to train a dog	59.0	41.0
Used an online or digital training programme	3.3	96.7
Other	8.6	91.4
None of these—my dog was already trained when I got him/her	9.0	91.0
None of these—I haven’t trained my dog in anyway	11.2	88.8

**Table 11 animals-14-00396-t011:** Behaviours displayed that respondents would like to change (*n* = 1814).

Answer	Frequency	%
Barking or vocalising for more than five minutes at a time when someone is present	197	10.9
Growling or snarling	108	6.0
Biting other dogs	44	2.4
Jumping up at people	404	22.3
Aggression towards people	50	2.8
Aggression towards other pets	150	8.3
Inappropriate toileting in the house	99	5.5
Inappropriate sexual mounting (“humping”)	85	4.7
Urinating when excited	78	4.3
Destructive behaviour	47	2.6
Signs of distress when left alone e.g., scratching, destructive behaviour, barking or howling for more than five minutes or toileting in the house	115	6.3
Not coming back when called	333	18.4
Chewing items not designed for chewing	126	6.9
Showing signs of fear	177	9.8
Other	123	6.8
None of these	652	35.9

## Data Availability

All data are owned by PDSA and have been shared with permission. Further information can be found here: https://www.pdsa.org.uk/media/3290/pdsa-paw-report-2017_online-3.pdf (accessed on 22 January 2024). Data can be made available by request to tipton.emma@pdsa.org.uk.

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
