# Peer review of "What Do We Know about Dog Owners? Exploring Associations between Pre-Purchase Behaviours, Knowledge and Understanding, Ownership Practices, and Dog Welfare"

_animals, 2024, doi:10.3390/ani14030396_

Round 1
Reviewer 1 Report
Comments and Suggestions for Authors
I am honored to be involved in this very interesting article. This study will be of interest to all researchers looking for concrete solutions to improve the welfare of captive animals in the future.
I have listed below the points that caught my attention. Please consider them.
1.
Although it is stated that exploring "ownership styles" is an important aim of this study, I feel that it is not mentioned much in the Results and Discussion section.
Should the PCA results discussed in the Methods section not be included in the Results section? In addition, should the discussion of the PCA results not be interpreted in the main body of the text, rather than a little mention in the Limitation? If not, please check the consistency with the Aim of the Introduction Section.
In addition, the connection between the Aim and the Research Questions is unclear. This may be because one of the two main objectives stated in the Aim, "exploring ownership styles," is not present in the Research Questions.
Furthermore, the connection between the second objective and the main theme of this study, "pre-purchase behaviours," is also not clear.
Please point out if my understanding is insufficient.
2.
I felt that the amount of text in the Introduction Section was a little too much.
I think the main element of this study is "pre-purchase behaviours". If so, I would suggest that you reduce the amount of text by merging paragraphs that refer to elements other than "pre-purchase behaviours".
This would better facilitate the understanding of the Aim and Research Questions.
Please consider this.
3.
In the conclusion section, I describe the study in more specific terms. This makes it easier to convey the considerations of this study.
4.
There is a problem with the numbers in the Subsection (e.g. 1.1) and SubSubsection (e.g. 1.1.1). Please check.
5.
Was the type or number of problem behaviors you focused on determined by reference to some scale or article?
6.
What are the specific contents of the dog training program in Table 5? Is there any concern that perceptions of this content may vary among respondents?
7.
Household size is not mentioned in Table 1. Is this an error?
8.
I think the specific content and quality of pre-purchase advice varies widely from one respondent to another. This is a very difficult issue, but I would like to hear the author's opinion on this point in future research.
Reviewer 2 Report
Comments and Suggestions for Authors
A paper that investigates the factors of owner knowledge and behaviours on welfare. The paper is well written and methods are clear and results reported appropriately. While this is more of an exploratory study the findings are important and useful to further our knowledge of owner behaviours.
However, I have a few concerns that impact on the interpretation of the results and that could be addressed to improve the work.
Line 198 please explain what is meant by participate a limited number of times. Is that in this survey or any survey?
Figure 1 and the subsequent reference does not match up with the standard BCS for dogs- there is no waist line in the considered "perfect" dog and the considered "thin" or "very thin" dogs do not appear to have obvious ribs and appear to be more equivalent to the standard ideal dog. More importantly the last image is the only image to have a tail up. Thus there may have been bias in the answers to this question. This should be addressed in the limitations and caution given with the interpretation of any dog body shape associations.
All subsections need to be numbered correctly.
Why were associations and two way comparisons used rather than factor analysis and regressions? Binary regressions, for example, could have explored the relevant associated factors taking all aspects into account and considering collinearity in the different factors.
Line 477 Further discussion could be given to- rather than speculating the possible bias explore the difference in questions between this and other studies.
Section from 490 further citations could have been included in the discussion to back up the speculation.
Round 2
Reviewer 1 Report
Comments and Suggestions for Authors
I have checked the new manuscripts.
Thank you for your response.